# Health Technology Assessment Report on Vagus Nerve Stimulation in Drug-Resistant Epilepsy

**DOI:** 10.3390/ijerph17176150

**Published:** 2020-08-24

**Authors:** Carlo Efisio Marras, Gabriella Colicchio, Luca De Palma, Alessandro De Benedictis, Giancarlo Di Gennaro, Marilou Cavaliere, Elisabetta Cesaroni, Alessandro Consales, Sofia Asioli, Massimo Caulo, Flavio Villani, Nelia Zamponi

**Affiliations:** 1Neurosurgery Unit, Department of Neuroscience, IRCCS Bambino Gesù Children Hospital, 00165 Rome, Italy; alessandro.debenedictis@opbg.net (A.D.B.); marilou.cavaliere@opbg.net (M.C.); 2Department of Neurosurgery, UCSC Gemelli University Hospital, 00167 Rome, Italy; colicchiogabriella@gmail.com; 3Pediatric Neurology Unit, Department of Neuroscience, IRCCS Bambino Gesù Children Hospital, 00165 Rome, Italy; luca.depalma@opbg.net; 4Neurology Unit, IRCCS Neuromed Institute, 86077 Pozzilli, Italy; gdigennaro@neuromed.it; 5Institute of Neurosurgery, University of Milan Bicocca, 20900 Milan, Italy; 6Pediatric Neurology Unit, Salesi Children Hospital, 60123 Ancona, Italy; elisabetta.cesaroni@ospedaliriuniti.marche.it (E.C.); nelia.zamponi@gmail.com (N.Z.); 7Pediatric Neurosurgery Unit, G. Gaslini Hospital, 16147 Genoa, Italy; alessandroconsales@gaslini.org; 8Department of Biomedical and Neuromotor Sciences, Section of Anatomic Pathology, Bellaria Hospital, University of Bologna, 40139 Bologna, Italy; sofia.asioli3@unibo.it; 9Department of Neuroscience, Imaging and Clinical Sciences, University of Chieti, 66100 Chieti, Italy; caulo@unich.it; 10Division of Clinical Neurophysiology and Epilepsy Center, IRCCS, San Martino Hospital, 16132 Genoa, Italy; flavio.villani@hsanmartino.it

**Keywords:** vagus nerve stimulation, health technology assessment, drug-resistant epilepsy

## Abstract

*Background*: Vagus nerve stimulation (VNS) is a palliative treatment for medical intractable epileptic syndromes not eligible for resective surgery. Health technology assessment (HTA) represents a modern approach to the analysis of technologies used for healthcare. The purpose of this study is to assess the clinical, organizational, financial, and economic impact of VNS therapy in drug-resistant epilepsies and to establish the congruity between costs incurred and health service reimbursement. *Methods*: The present study used an HTA approach. It is based on an extensive detailed bibliographic search on databases (Medline, Pubmed, Embase and Cochrane, sites of scientific societies and institutional sites). The HTA study includes the following issues: (a) social impact and costs of the disease; (b) VNS eligibility and clinical results; (c) quality of life (QoL) after VNS therapy; (d) economic impact and productivity regained after VNS; and (e) costs of VNS. Results: Literature data indicate VNS as an effective treatment with a potential positive impact on social aspects and on quality of life. The diagnosis-related group (DRG) financing, both on national and regional levels, does not cover the cost of the medical device. There was an evident insufficient coverage of the DRG compared to the full cost of implanting the device. *Conclusions*: VNS is a palliative treatment for reducing seizure frequency and intensity. Despite its economic cost, VNS should improve patients’ quality of life and reduce care needs.

## 1. Background

Epilepsy is one of the most common neurological diseases [1]. According to World Health Organization data (WHO) [2], epilepsy affects about 50 million people worldwide, with approximately 6 million in Europe, and around 500,000 in Italy. Every year, more than 80 people per 100,000 receive a diagnosis of epilepsy, most commonly during adolescence. Epilepsy represents 0.5% of the global burden of disease and has significant economic implications in terms of healthcare needs, welfare, premature death, and work productivity loss [3]. Social stigma that surround this condition worldwide are often more difficult to overcome than seizure itself [2,4].

Resective surgery is effective in selected patients with focal drug-resistant epilepsy. The results collected in both adults and children show a seizure freedom ranging between 40% and 90% [5,6,7,8,9]. In 1997 the Food and Drug Administration (FDA) and the agencies of the European and Canadian registration approved VNS as an additional therapy to reduce seizure frequency in adults, adolescents and children whose seizures are refractory to antiseizure drugs (ASDs) and who are not eligible for resective surgery. VNS is a neuromodulation approach that uses a surgically implantable, programmable pulse generator powered by a battery connected to a helical bipolar lead. The lead is attached to the mid cervical portion of the left vagus nerve and delivers a biphasic current that continuously cycles between on and off periods [10,11,12,13]. Studies of VNS efficacy showed, in up to 60% of implanted patients, a seizure frequency reduction higher than 50% both in adults and pediatric population [14,15,16,17,18,19,20]. The decrease in seizure frequency results in a reduction in hospitalization rate, recovery time and emergency hospital admissions, and consequentially in a reduction in hospital, health and social costs [21,22,23,24].

Health technology assessment (HTA) is the bridge between science that produces evidence and decisions that may derive from evidence at different levels of healthcare system provision [5,25]. The concept of health technology is wide and covers the entire healthcare process, including diagnosis and treatments. Evidence can impact safety and efficacy as well as the organizational, social and ethical aspects of health technology [26]. Decision-making occurs on three levels. The macro level involves policy decisions. The meso level involves decisions by the regions or individual institutions (hospitals and health trusts). The micro level involves clinical practice [27]. If the evidence to produce is organizational–managerial and the decisions to take are at micro level as they concern clinical practice, decision-makers can benefit from a process that pinpoints diagnostic–therapeutic work-up and quantifies the relative costs. The purpose of this HTA report is to assess the clinical, organizational, financial and economic impact of managing of patients with drug-resistant epilepsy. The main aim of the report is to establish the congruity between costs incurred and health service reimbursement for the management of patients with drug-resistant epilepsy treated with VNS therapy.

## 2. Methods

The present report is based on an extensive detailed bibliographic search on databases, such as Medline, Pubmed, Embase and Cochrane; sites of scientific societies; sites of institutions, such as the Italian Ministry of Health, FDA—Food & Drug Administration, and NICE—National Institute for Clinical Excellence; institutional sites of international HTA agencies: International Network of Agencies for Health Technology Assessment (INAHTA), Canadian Agencies for Drugs and Technologies in Health (CADTH), and National Coordinating Center for Health Technology Assessment (NCCHTA)—administration regions; and gazettes. Analysis related to costs is crucial to the economic assessment and it may turn out to be a useful tool to solve specific problems when choosing among available alternatives. The analysis of relevant literature was conducted using the following key words: “HTA”, “VNS”, “VNS costs”, “Epilepsy costs”, “HTA Epilepsy”, “Epilepsy social impact”. The rough selection using the terms HTA and VNS included a huge number of articles. Then, matching “HTA” with “VNS”, “Epilepsy”, and “Epilepsy costs”, the number of papers reduced substantially from 53686 (HTA only) to 303. Most of the 303 papers selected were not focused on this issue, and very rarely analyzed epilepsy surgery; therefore, 17 contributions were considered eligible for this study (Figure 1).

By matching the term “VNS” with the issues of this HTA study (“social impact and costs”, “eligibility and clinical results”, “quality of life”, “economic impact and productivity”), the number of papers decreased from 1939 to 95 because of some constrains including epilepsy, year of publication (from 2000) and studies including at least 10 cases (Figure 2).

Analysis related to costs is crucial to the economic assessment, and it may turn out to be a useful tool to solve specific problems when choosing among available alternatives. The pillars of the present evaluation are cost (or opportunity cost) and benefit, i.e., the advantages or positive consequences of the action in question. Economic evaluation was performed considering the data of unit costs presented in “HTA Report on the management of drug-resistant epilepsies” of LICE (Lega Italiana Contro l’Epilessia) [28]. Data from three Italian hospitals in three different regions (Lazio, Emilia Romagna and Lombardia) were used to calculate the national average cost. The cost was calculated by averaging the unit price of the individual materials used for the treatment.

This HTA analysis focused mainly on the following issues: (a) social impact and costs of the disease; (b) clinical results after VNS therapy; (c) quality of life after VNS therapy; (d) economic impact and productivity regained after VNS; and (e) costs of VNS.

## 3. Results and Discussion 

### 3.1. Social Impact and Costs of the Disease

The social burden, in terms of stigma and poor quality of life in patients of different ages, prognosis, comorbidity, and treatment response, has to be considered in the evaluation of the epilepsy’s global burden [2]. In general, the concept of epilepsy as mental disorder is no longer accepted worldwide; however, epilepsy patients are often dealt with by society with stereotypical attitudes, such as fear or suspicion, as if they had psychological problems or mental retardation, or were considered to be unreliable in work activities due to seizures [29]. Epilepsy is a social issue even among the pediatric population—the Adolescent Mental Health Survey reported that approximately 25% of children aged 5–14 years had psychiatric difficulties with respect to 9% of controls; children with epilepsy and psychiatric difficulties also had neurological comorbidities. Overall, children with epilepsy are growing with significant social problems that include less opportunities for future employment, lower chances of getting married, and possible difficulties in social relationships and in having an independent life [30,31]. A review with economic modeling has estimated the cost of epilepsy in 28 European countries. Despite a prevalence of 4.3 to 7.8 patients per 1000 persons, the total cost in Europe was estimated at EUR 15.5 billion, of which the indirect costs accounted for 55%, the direct costs of health (particularly outpatient care which entailed an expenditure of EUR 1.3 billion) accounted for 18%, and the non-medical cost for 27% [32]; the cost per case treated/year ranged from EUR 2000 to 11,000. The economic burden of epilepsy is substantial, and it is inversely proportional to seizure control. Costs are higher in the first year after diagnosis than in the following years and vary according to the age of the subject [33,34]. The average cost per patient per year is higher for children than for adults and for newly diagnosed patients for whom the first diagnosis of epilepsy is addressed at the first visit. The major cost driver is hospitalization (63.7%), followed by drugs (10.5%), day-hospital visits (4.1%), outpatient visits (3.85%), other tests (3.1%), and electroencephalograms (2.3%) [35]. In particular, direct costs (outpatient and hospital) are based also on the age of onset of the disease, epilepsy features, frequency of seizures, and type of ASDs taken. Direct costs are higher for children and for the elderly over the age of 60, and they decrease during the second year of treatment. In addition, indirect costs (for example: lost productivity) account for about half of the total costs. Among the direct costs of healthcare, outpatient costs are prevalent in more stable patients, whereas hospital costs are prevalent in patients with higher frequency of seizures [23,36,37,38,39].

### 3.2. VNS Eligibility and Clinical Results

VNS has been used as an adjunctive treatment for drug-resistant epilepsy for more than 25 years and it seems to be effective in different types of epilepsy. VNS is a surgical procedure that may be indicated in drug-resistant patients excluded from resective surgery. The mechanism of action of this intermittent electrical stimulation of the vagus nerve is not completely understood [11]. The evaluation of the clinical result classifies a “responder” as a patient experiencing at least a 50% of seizures reduction [40,41] (Table 1). In order to evaluate the efficacy and safety of VNS, the American Academy of Neurology (AAN) Guideline Development Subcommittee reviewed the full text of 216 articles published within 17 years. Just one article of the whole revision had an evidence level higher than class III [40]. The results collected had a level C of recommendation: (a) VNS as adjunctive treatment for children with partial or generalized epilepsy; (b) VNS potentially useful in patients with Lennox–Gastaut syndrome (LGS); (c) VNS progressively effective in patients over multiple years of exposure; and (d) VNS magnet activation associated with seizure abortion when used at the time of seizure auras [40]. Literature data from clinical trials on VNS established the following evidence: VNS reduces by at least 50% the frequency of seizures in 21–75% of subjects; the benefit of treatment might persist longer than 15 years of follow-up; and both adults and children could benefit from the treatment in 50–62% of patients [16,17,19,23,42,43,44,45,46,47,48,49]. On the other hand, a lower rate of responders is reported in other studies [15,50,51,52,53,54,55,56,57,58]. Among the huge bibliography reporting the outcome, numerous reviews remark not only the rate of seizure reduction but also the progression of the efficacy over years. Orosz et al. analyzed 347 children after VNS implantation. At 6, 12, and 24 months after implantation, 32.5%, 37.6%, and 43.8%, respectively, of patients had a ≥50% reduction in baseline seizure frequency of the predominant seizure type [59]. Favorable results were also evident for all secondary outcome measures, including changes in seizure duration, ictal and postictal severity, quality of life, clinical global impression of improvement, and safety [57]. The non-responders rate ranges between 25% and 65%, and it could be useful to better define selection criteria for candidates to VNS. The VNS initial indication was restricted to drug-resistant patients, was not suitable for resective surgery, and was affected by partial epileptic seizures [17]. The Cochrane report concluded that VNS appears to be an effective and well-tolerated treatment for partial seizures; at the time of publication, however, VNSs were utilized in all ages and different kinds of epilepsies, syndromes and etiologies.

VNS is a neuromodulation approach that, among the palliative surgical therapies, is the most well-used worldwide. Since 1994, the first VNS study group concluded a safe and tolerable therapy for the implantation procedure, stimulating system and patient acceptance [13]. Kavcic et al. point out adverse events: hoarseness, increase salivation, cough, snoring, cervical muscle spasms and laryngeal pain. All effects, except for hoarseness during stimulation (serving as a marker of device function), were much milder or not present at the early follow-up visits [48]. Orosz et al. report complications related to surgery and to device malfunction ranging from 2 to 5%, all reversible and in line with other literature data [16,40,59,60].

Although more than 100,000 epileptic patients underwent VNS, the predictors of outcome are still under investigation, and the data reported are contradictory [43]. The efficacy of VNS on epilepsy at a younger age is still a matter of discussion [40,45,46,48,60,61,62,63]. Shorter history of epilepsy as a positive factor, reported by Kavcic, Wang and some other authors [48,64,65,66], is not accepted by all [61,63,67]. VNS appeared to have a role in malformation of cortical development, tuberous sclerosis, and post-traumatic and post-stroke epilepsies [46,68]. Localization-related epilepsies [69,70] and patients with interictal EEG focal activities have a better clinical outcome [68,71], but some authors report positive results in multifocal EEG patients as well as in generalized epilepsies [43,46,62,71,72,73,74,75,76,77,78]. Severe syndromes, such as Lennox–Gastaut and Dravet, show a high responder rate [20,79,80,81]. VNS has been used in very severe conditions including epilepsia partialis continua (EPC) and status epilepticus (SE). De Benedictis et.al report a series of four patients with SE who saw an improvement in seizure after 5 weeks since stimulation onset, and a persistence of the effect after a mean follow-up of 3 years [82]. On the other hand, a review of 17 papers did not recommend the use of VNS for resistant status epilepticus (RSE); nevertheless, a more recent review by Dibué-Adjei concludes that VNS can interrupt RSE in 74% of patients, with a class IV evidence level [83]. VNS was considered as a promising tool in the prevention or reduction in sudden unexpected death in epilepsy (SUDEP) [84,85,86]. However, the audit of 466 VNS cases enrolled at a single center showed that patients with VNS carried a similar risk for SUDEP as other drug refractory patients [87]. Tomson and Ryvlin suggested that the risk of SUDEP significantly decreases during long-term follow-up in VNS therapy [88,89].

### 3.3. Quality of Life (QoL), Quality Adjusted Life Years (QALY) and Health-Related Quality of Life (HRQOL) after VNS

The PuLsE (open prospective randomized long-term effectiveness) trial demonstrated that VNS therapy, adjunct to the best medical practice (BMP) in patients with drug-resistant focal seizures, was associated with a significant improvement in quality of life compared with BMP alone [94]. These data are widely confirmed by the literature [18,90,91,94,95,96,97,98,99]. The main findings after VNS therapy are the following: reduction in seizure frequency; improvement in verbal and figure recognition; increased experiences of attention and arousal; memory consolidation; cognitive flexibility and creativity; and decision-making [92,93,100,101,102]. However, changes are not significant or cover a limited number of domains only, especially in studies based on small samples [103]. The benefits are also quantified in terms of quality adjusted life years (QALYs), where the preference for a health state is quantified as a utility value, and then multiplied by years spent in the health state or years expected to live in a similar way [104,105]. QALY is an indicator that takes into account both the quality and the quantity of life lived. Helmers et al. study reports the QALY gained over lifetime after VNS implantation among pediatric patients. This study demonstrates that children gain about six QALYs and adolescents gain about 5 QALYs on average over the entire duration of life. The impact of VNS on QALY was assessed taking into account the following: the difference in the average utility score of QoL between the post-VNS period and the 6-months pre-VNS period; age; gender; and life expectancy in the entire VNS population [106]. VNS improved different health-related quality of life (HRQOL) domains—mainly cognitive, psychological, and social [92,107,108,109]—but it also improved memory, mood, behavior, alertness, achievement, and verbal skills [97,98,110,111,112,113].

### 3.4. Economic Impact and Productivity Regained after VNS

European Economic studies have shown that VNS treatment is beneficial from the perspective of third party payers and cost-effective from a societal perspective [114]. In a pilot study of 20 drug-resistant patients who were not candidates for surgical resection, VNS improved outcomes by reducing the direct cost of healthcare (drugs, outpatient consultations, hospitalizations, lab tests, device insertion, etc.) by about USD 3000 per year on average [115]. In fact, the total direct cost per patient pre- and post-implantation was approximately USD 6700 and UDS 3600, respectively [116]. A significant decrease in the direct cost of VNS-treated patients compared with conservatively treated patients was also reported by other groups [22,23,117,118,119,120].

Ben-Menachem et al. reported a saving in hospital costs by VNS equal to USD 3000 per patient per year by analyzing the direct costs of hospitalization in 43 drug-resistant patients in the 18 months preceding and 18 months following the implantation of the device [121]. The hospitalization costs were reduced both in VNS responders (reduction in seizures >25%) and in non-responders (reduction in seizures <25%). In the latter group, the intensive care unit (ICU) admissions were reduced from 6 to 0 with an average hospital stay from 24 to 0 days. In addition, emergency room admissions were reduced by 50% (from 8 to 4) and in-hospital admissions from 16 to 5, with a reduction in days of hospitalization from 122 to 28. Consequently, the costs of intensive care were cancelled, whereas those related to the emergency room and in-hospital stay significantly reduced.

The use of the VNS device has a positive effect on the consumption of healthcare resources and therefore on reducing the operating costs of the drug-resistant patients [18,122,123]. Data collected from 321 patients one year before to three years after VNS implantation showed a reduction higher than 70% of accidents and emergency unit attendance, as well as a decrease in severe injuries. The study also reported a decrease in elective (7%) and non-elective (14%) inpatient episodes [124]. Comparable results were collected in a larger group of patients [125]. Indeed, Helmers et al. showed that for patients with an age range of 1–11 years (*n* =238), hospitalizations and emergency room admissions were reduced post-VNS vs. pre-VNS (adjusted Internal Rate of Return (IRR) = 0.73 and 0.74, respectively). Average total healthcare costs were lower post-VNS vs. pre-VNS (USD 18,437 vs. USD18,839 quarterly). Additionally, for 12–17 years old patients (*n* = 207), hospitalizations and episodes of status epilepticus were reduced post-VNS vs. pre-VNS (adjusted IRR = 0.43 and 0.25, respectively). Average total healthcare costs were lower in the post-VNS vs. the pre-VNS period (USD 14,546 vs. USD19,695 quarterly) [106].

In a previous study, Helmers et al. demonstrated that resource use and epilepsy-related events gradually decreased after stimulator implantation [126]. As a matter of fact, VNS has a, low rate of complications and reoperations with a 2% incidence, and it is also associated with reduced seizure-related emergencies department visits and hospital admissions [125,127].

Finally an American study on VNS estimated the time spent on the management of the disease and consequently the time lost from work in 138 patients during the first year after VNS implantation [122]. The number of working days lost because of the disease reduced from 3.67 to 1.04 days, and the average time spent on the management of the disease decreased from 352.6 to 136.1 min per week. The economic value of the recovery of 2.63 days of average productivity per patient following implantation of VNS generates an economic return of EUR 275 per patient. This evaluation is the average of gross domestic product (GDP) per capita (resulting from the total yearly amount of USD 22,995 estimated by the International Monetary Fund, then divided by 220 working days).

### 3.5. Costs of VNS

Based on the calculations performed, the average total cost at the national level in Italy for the VNS implantation stage is EUR 26,543 (Table 2), and it includes the following: the cost of the VNS device, which is fixed and equal to EUR 21,084; the cost of surgery, which is EUR 3518 on average; and the cost of hospitalization, amounting on average to EUR 1941.

Considering the cost of surgery, the variability among the different centers involved in the study derives from several factors (Table 3). Among these, the average duration of surgery, for example, ranging between one to three hours, accounts for the duration of the occupation of the operating room that could last between a minimum of 2 to a maximum of 5 h, therefore resulting in an increase in the cost of the healthcare personnel employed during surgery. Finally, the variability of days of bed occupancy and the days of hospital stay have to be included.

In Table 4, the costs of each phase of VNS implantation are reported. For each phase the higher, lower, and average cost on the national level are considered.

By excluding the cost of the device and of preoperative tests from the calculation of the total cost of the VNS implantation stage, the surgery phase accounts for 64% of the average total cost, while the phase of hospitalization account for the remaining 36%. However, the cost of the device accounted for 79% of the total cost, the cost of the surgery phase accounted for 13%, and the cost of the hospitalization phase accounted for 7%. Moreover, in good responder patients, after a period ranging between 5 and 7 years, a new internal pulse generator has to be replaced with additional costs of surgery and hospitalization, thus incurring a supplementary cost of EUR 16,000.

Regarding the remuneration of the Italian Health National System, the procedure of VNS implantation is classified in the category DRG 008 (operations on cranial and peripheral nerves and other intervention on the nervous system without complications).

Following the Italian Ministerial Decree published in 2006, a reimbursement of EUR 2770 per hospitalization is provided, and this amount was updated in 2012 to EUR 2326. However, the DRG financing does not cover, both on national and regional levels, the cost of the medical device which amounts to an average of EUR 21,000 (national price without VAT (value added tax) provided by the global medical technology company LivaNova. In order to cover the cost of the devices, in the last ten years, an additional reimbursement was provided in an increasing number of Italian regions. The reimbursement paid at present is insufficient even if the extra rate is added to the DRG 008 rate specifically for each region, amounting to EUR 15,000 for the Emilia Romagna and to EUR 13,312 for Lombardia. Currently, 9 out of 20 regions benefit from an extra payment. Although the reimbursement progressively improved for more 50% of the Italian population, an evident insufficient profitability of the DRG compared to the full cost of the implant device is still present.

## 4. Conclusions

In conclusion, our analysis indicates that VNS is a palliative treatment for reducing seizure frequency and intensity in drug-resistant epilepsy patients not eligible for resective surgery. Although large numbers of patients have been implanted worldwide, the amount of evidence on the eligibility criteria for VNS remains insufficient. Both responsive patients and their care-takers experience a subjective improvement in their quality of life. Despite its economic cost, VNS reduces care needs; therefore, its use may be justified in responder patients. There is an insufficient coverage of the DRG-based reimbursement compared to the full cost of the implant device.

## Figures and Tables

**Figure 1 ijerph-17-06150-f001:**
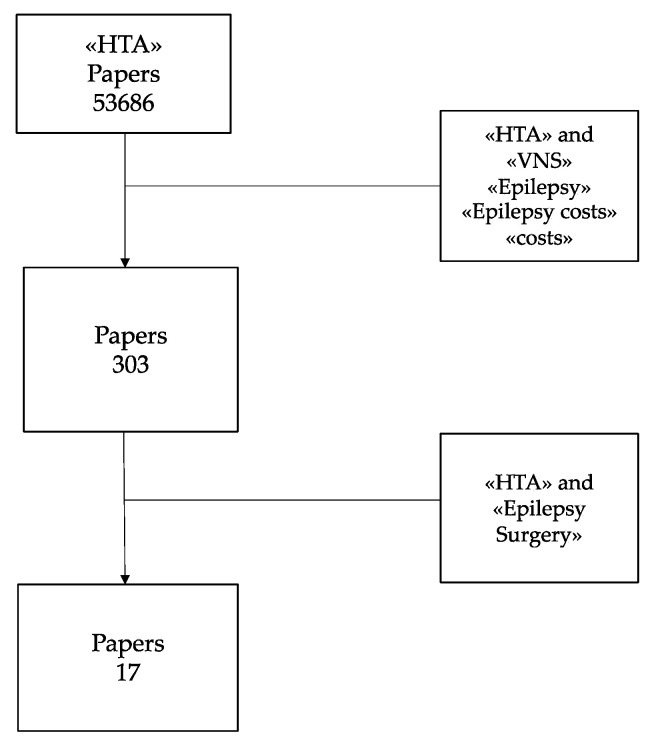
Literature search progression: (Health Technology Assessment: HTA).

**Figure 2 ijerph-17-06150-f002:**
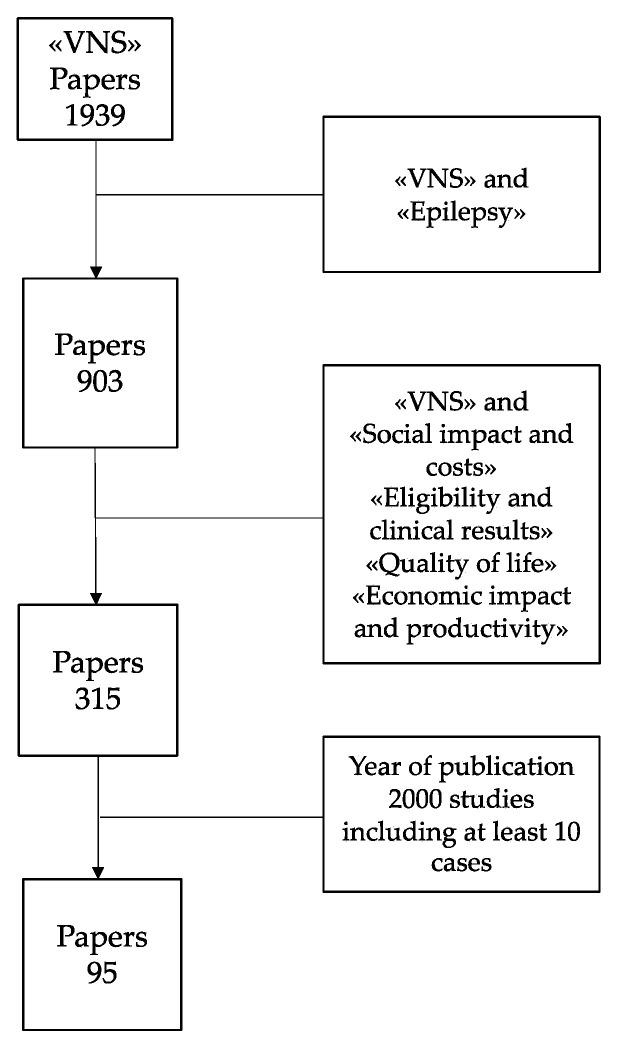
Literature search progression.

**Table 1 ijerph-17-06150-t001:** VNS series including a minimum 10 patients with a follow-up higher than 6 months.

Author (Ref)	Patients Number	Mean/Range Age at Implantation	Mean/Range Follow-Up	Seizure Free % (*n*)	Responders % (*n*) of Patients with ≥50% Seizure Reduction
Abubakr [15]	31	14 to 62 years	4 years	-	53% (16)
Gonzales [19]	1194	n.d.	24–48 months	8%	63%
Morris [40]	481	n.d.	1–5 years	n.d.	55%
Parain [42]	10	5 to 20 y	22 months	0%	90%
Janszky [43]	47	22.7 y	22 months	13%	n.d.
Klinkenberg [45]	41	4 to 18 y	10 months	n.d.	26%
Englot [46]	3321	n.d.	3 to 64 months	n.d.	21 to 50%
Englot [47]	4483 1104	-	3 months 24 months	-	44% 56%
Chrastina [49]	74	18–59 years	10 years	n.d.	63.6%
Kuba [50]	90	36.3 years	5 years	(5)	(44/85)
Vonck [51]	131	32 years	33 months	7%	50%
De Herdt [52]	138	30 years	44 months	9% (12)	13%
Patwardhan [54]	38	11 months–16 years	12 moths (10 months–18 months)	n.d. (*n* 29% >90% reduction)	68%
Nakken [55]	47	34.4 years	2.7 years	2% (1 pt)	32% (15)
Fernandez [58]	15	3 years <	4.3 years (1.4 years–10.2)	0%	categorical variable reported 33% “improvement”
Terra [57]	36	up to 18years	12 months to 4 years	(1 pt)	55% ≥ 50% seizure reduction
Orosz [59]	347	6 months to 17.9 years	24 months	8% (17/208)	43.8%
Kabir [60]	69	10.69 years	6 months to 10 years	1.4% (1)	8.7% (6) Engel II 90% (63) Engel III–IV
Alexopoulos [61]	46	12 years	2 years	10% (5)	43.5% (20)
Colicchio [62]	135	5 months to 64 years	36 months	5.1%	49%
Wang [66]	1061	5–60 years	6 months–12 years	n.d.	53.53%
Ghaemi [68]	144	3 to 65 years	2 years	(10)	62% (89)
Marras [69]	35	6 to 52 years	36 months	-	51% (18)
Amar [90]	921 CS 3822 Non-CS	28 years 26 years	24 months 24 months	5% 8%	55% 62%
Ulate-Campos [91]	30	21 months (1–144)	6–36 months	n.d.	50%
Moro de Faes [92]	35	3–18 years	3 months to 2–3 years	n.d.	43%
Soleman [93]	45	133.9 ± 184.5 months	72.3 ± 39.8 months	n.d.	49.9%

Legend: underwent cranial surgery (CS) for epilepsy; not described (nd); patient (pt).

**Table 2 ijerph-17-06150-t002:** Full cost hospital of VNS implantation—region details.

Items	Lazio	Emilia Romagna	Lombardia	National Average
Surgery	EUR 3182	EUR 5168	EUR 2204	EUR 3518
VNS device	EUR 21,084	EUR 21,084	EUR 21,084	EUR 21,084
Hospital stay	EUR 1327	EUR 2293	EUR 2203	EUR 1941
Total costs	EUR 25,593	EUR 28,545	EUR 25,491	EUR 26,543
Total costs without device	EUR 4509	EUR 7461	EUR 4407	EUR 5459

**Table 3 ijerph-17-06150-t003:** Consumption of health resources in VNS implantation stage.

Health Resources	Min	Max	National Average
Days of hospitalization/patient (no.)	3	4	4
Average duration of surgery (hours)	1	3	2
Duration occupancy operating room (hours)	2	5	3
Hours of intensive care/patient (no.)	0	0	0

**Table 4 ijerph-17-06150-t004:** Full average cost of VNS implantation stage.

Items	Min (EUR)	Max (EUR)	National Average (EUR)
Pre-operative diagnostic tests	380	380	380
Surgery	-	-	-
Materials	38	80	65
Personnel costs	293	931	638
Drugs	189	199	193
Operating room	1684	3958	2622
VNS device	21,084	21,084	21,084
Total cost of intervention	23,288	26,252	24,602
Total cost of intervention without VNS device	*2204*	*5168*	3518
Inpatient	-	-	-
Hospital stay	1067	1869	1572
Inpatient drugs	-	5	3
Personnel costs for inpatient	233	392	339
Other (e.g., perfusion)	27	27	27
Total cost of inpatient	1327	2293	1941
Total cost of VNS implantation stage without pre-operative tests	24,615	28,545	26,543
TOTAL COST OF VNS IMPLANT	24,995	28,925	26,923

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
