# Peer review of "Health Technology Assessment Report on Vagus Nerve Stimulation in Drug-Resistant Epilepsy"

_ijerph, 2020, doi:10.3390/ijerph17176150_

Round 1

Reviewer 1 Report

The submitted manuscript, dealing with the HTA analysis on VNS treatment contributes to establishing the cost-effectiveness of the procedure in patients who have drug-resistant epilepsy. The authors formulated five research questions, which correlates well with the purpose of the assessment. Data demonstrated herein are relevant to the questions. However, several improvements are recommended, as listed below, to make this paper more impactful.

  1. First of all, it is recommended to provide a more rigorous description of the search strategy and inclusion and exclusion criteria for data extraction, which should also be presented in the form of a flow diagram of the literature search progression.
  2. To support the cost-effectiveness of VNS treatment, it is recommended to provide also data on safety analysis.
  3. Regarding clinical effectiveness, there is a lack of evidence supporting the beneficial effect of VNS treatment on seizure intensity indicated in conclusions.
  4. Lines 146-147, it is recommended to include in Table 1 all cited references demonstrating clinical evidence of the effects of VNS on seizure frequency and rate of responders.
  5. The indicated in line 241, the average total cost of the VNS implantation stage at the national level should be put in Table 2.

Author Response

Here are reported point by point our reply to the reviewer

1. First of all, it is recommended to provide a more rigorous description of the search strategy and inclusion and exclusion criteria for data extraction, which should also be presented in the form of a flow diagram of the literature search progression.

@ A description of the search strategy was expanded and included from the line 87 (line 96 of the reviewed paper):

Analysis related to costs is crucial to the economic assessment and it may turn out to be a useful tool to solve specific problems when choosing among available alternatives. The analysis of relevant literature was conducted using the following key words: "HTA", "VNS", "VNS costs", "Epilepsy costs", "HTA Epilepsy", "Epilepsy social impact". The rough selection using the terms HTA and VNS included a huge number of articles. Then matching “HTA” with “VNS”, “Epilepsy”, “Epilepsy costs” the number of papers reduced substantially from 53686 (HTA only) to 303. Most of the 303 papers selected were not focused on this issue, and very rarely analyzed epilepsy surgery: therefore 17 contributions were considered eligible for this study. Matching the term “VNS” with the issues of this HTA study (“social impact and costs”, “eligibility and clinical results”, “quality of life”, “Economic Impact and productivity”) the number of papers decreased from 1939 to 95 because of some constrains including epilepsy, the year of publication (from 2000) and studies including at least 10 cases.”

Two diagrams are also reported respectively on line 105 and on line 111 of the reviewed manuscript

2. To support the cost-effectiveness of VNS treatment, it is recommended to provide also data on safety analysis.

@ According to the question, we added on lines 158 and 230 some data on safety analysis (lines 184 and 264 of the revised paper):

Line 158: “VNS is a neuromodulation approach that, among the palliative surgical therapies, is the most diffuse worldwide. Since 1994 the VNS first study group concluded for a safe and tolerable therapy for the implantation procedure, stimulating system and patient acceptance.  Kavcic et al point out on adverse events: hoarseness, increase salivation, cough, snoring, cervical muscle spasms and laryngeal pain. All effects, except for hoarseness during the stimulation (serving as a marker of device function) were much milder or gone at the early follow up visits. Orosz et al. report   complications related to surgery and to device malfunction ranging from 2 to 5%, all reversible and in line with other literature data.”

Line 230:As a matter of fact VNS has low rate of complications and reoperations with a 2% incidence, and it is also associated with reduced seizure-related emergencies department visits and hospital admissions.”

@ a new reference was also included: Revesz D,Rydenhag B, Ben-Menachem E. Complications and safety of vagus nerve stimulation: 25 years experience at a single center.J Neurosurg Pediatr 2016 18:97-104. http://thejns.org /doi/abs/10.3171/2016.1.PEDS1553

3. Regarding clinical effectiveness, there is a lack of evidence supporting the beneficial effect of VNS treatment on seizure intensity indicated in conclusions.

@ The assumption of lack of evidence of the efficacy of VNS on epilepsy is correct considering the whole population of the patient treated. However, the responder group benefits VNS in terms of the issues discussed, including quality of life. So far in the conclusion (abstract included), we specify VNS as a palliative treatment and we underline its efficacy only in the responder group.

4. Lines 146-147, it is recommended to include in Table 1 all cited references demonstrating clinical evidence of the effects of VNS on seizure frequency and rate of responders.

@ table 1 was expanded including the other series with at least 10 cases (in the previous table the series had at least 30 patients). Table 1 was then described as follows: VNS series including minimum 10 patients with a follow up higher than 6 months

5. As indicated in line 241, the average total cost of the VNS implantation stage at the national level should be put in Table 2.

@ done

Reviewer 2 Report

It is comprehensive paper. I have only minor comments regarding especially abbrevations. The authors use a lot of them which makes draft difficult to read, and they additionally are not consisent in using them (e.g. VNS, FDA, LGS, or SE). Please check them properly and remove those which are not neccessary. What is ICU (line 212)? Explain. Some typos: line 280.

Additionally, the name antiepileptic drugs AEDs is no longer proper. Since 2017, the recommendation is to use term „antiseizure drugs” with abb. ASDs. Indeed, thise drugs decrease number of seizures but do NOT prevent process of epileptogenesis. Adjust it in the whole draft. Morover, authors write the prevelance of epilepsy is estimated on 0.5% which is actually to small, its rather 1%. Change it please. Thank you.

Author Response

Here are reported the answers to the reviewer

1. I have only minor comments regarding especially abbrevations. The authors use a lot of them which makes draft difficult to read, and they additionally are not consisent in using them (e.g. VNS, FDA, LGS, or SE). Please check them properly and remove those which are not neccessary. What is ICU (line 212)? Explain. Some typos: line 280.

@ according to the editor indications we explain on the text all the abbreviations. Moreover after the keywords (line 46), we made a abbreviation section following alphabetic order (including ICU: Intensive Care Unit)

2. Additionally, the name antiepileptic drugs AEDs is no longer proper. Since 2017, the recommendation is to use term „antiseizure drugs” with abb. ASDs. Indeed, thise drugs decrease number of seizures but do NOT prevent process of epileptogenesis. Adjust it in the whole draft.

@ done

3. Morover, authors write the prevelance of epilepsy is estimated on 0.5% which is actually to small, its rather 1%. Change it please. Thank you.

The prevalence of epilepsy worldwide is very difficult to define. We believe that it is even higher than 2%. The paper we cited report 0.5%, and we appreciate if the reviewer should indicate the publication that reports a 1% of incidence. We should include both data and references.

Reviewer 3 Report

This study assessed the clinical, organizational, financial, and economic impact of Vagus Nerve Stimulation in Drug-Resistant Epilepsy. The introduction provides sufficient background and includes proper relevant references. Most of the methods were adequately described, the conclusion is well supported by the results. In the result section, more tables or figures are needed to better present the results. For example, pie chart should be used to present the percentage. Discussion section discusses related issues properly. All in all, the assessment is comprehensive, appropriate, and systematic. The manuscript has very good wording and logic. However, several papers in the similar topic have been presented (DOI: 10.1136/practneurol-2019-002210 or DOI: 10.1007/978-3-319-39546-3_8), which reduces the initiatives of the manuscript. Although this manuscript used a systematic HTA method, some of the assessment was covered in the previous paper.

Small err found:

line 33, clinical, organizational, financial, and economic impact

line 51, According to World Health 49 Organization data (WHO), but it still needs a reference here.

Author Response

Here are reported the answers to the reviewer

1. In the result section, more tables or figures are needed to better present the results. For example, pie chart should be used to present the percentage.

@ to make easier the paper reading, we enlarged table 1, and added two diagrams that allow a better understanding of the literature data selection (see pages 3 and 6 of the reviewed paper).

2. line 33, clinical, organizational, financial, and economic impact

@ done

3. line 51,According to World Health 49 Organization data (WHO), but it still needs a reference here

@ Pahl K, de Boer HM. Epilepsy and rights. In Atlas: epilepsy care in the world. Geneva: WHO 2005; p. 72-3.

Round 2

Reviewer 1 Report

N/A

Author Response

The reviewer doesn't ask further revisions

Reviewer 3 Report

The manuscript has been improved and now is good for publication in the IJERPH.

Please make sure "in line 33, clinical, organizational, financial, and economic impact" was done. It was not corrected in the version I downloaded.

-Additional comments:

The manuscript has been improved dramatically after the revision. This study assessed the clinical, organizational, financial, and economic impact of Vagus Nerve Stimulation in Drug-Resistant Epilepsy. The introduction provides sufficient background and includes proper relevant references. Most of the methods were adequately described, the conclusion is well supported by the results.  Discussion section discusses related issues properly. All in all, the assessment is comprehensive, appropriate, and systematic. The manuscript has very good wording and logic. I recommend the acceptance of this manuscript in the present version. 

Author Response

I apologize but it is not clear what revision is requested on the line 33.

Anyway a comma was typed before "and"